# Effect of 2 Years of Monthly Calcifediol Administration in Postmenopausal Women with Vitamin D Insufficiency

**DOI:** 10.3390/nu16111754

**Published:** 2024-06-03

**Authors:** Marco Occhiuto, Jessica Pepe, Luciano Colangelo, Marco Lucarelli, Antonio Angeloni, Luciano Nieddu, Viviana De Martino, Salvatore Minisola, Cristiana Cipriani

**Affiliations:** 1Department of Clinical, Internal, Anaesthesiologic and Cardiovascular Sciences, Sapienza University of Rome, Viale del Policlinico 155, 00161 Rome, Italy; marco.occhiuto@uniroma1.it (M.O.); jessica.pepe@uniroma1.it (J.P.); luciano.colangelo@uniroma1.it (L.C.); viviana.demartino@uniroma1.it (V.D.M.); cristiana.cipriani@gmail.com (C.C.); 2Department of Experimental Medicine, Sapienza University of Rome, Viale del Policlinico 155, 00161 Rome, Italy; marco.lucarelli@uniroma1.it (M.L.); antonio.angeloni@uniroma1.it (A.A.); 3Department of Humanistic and Social International Sciences, UNINT University, Via Cristoforo Colombo 200, 00147 Rome, Italy; l.nieddu@gmail.com

**Keywords:** calcifediol, vitamin D, ionised calcium, bone alkaline phosphatase

## Abstract

Background: We assessed the long-term (24 months) efficacy and safety of monthly calcifediol (0.266 mg) in the correction and maintenance of total 25(OH)D levels in postmenopausal women with basal values <30 ng/mL. Methods: We initially enrolled 45 consecutive patients during the period September 2019–September 2020. After an initial visit, patients were instructed to return at 3, 6, 9, 12 and 24 months for measuring serum total 25(OH)D, ionised calcium, creatinine and isoenzyme of alkaline phosphatase (bALP). Here, we report only the per-protocol analysis, because the COVID-19 pandemic precluded adherence to the scheduled visits for some patients. Results: The patients’ mean age was 62.4 ± 9.0 years. Mean basal 25(OH)D levels were 20.5 ± 5.3 ng/mL. There was a continuous increase of mean 25(OH)D values (*p* for trend < 0.001). However, mean values at month 24 (36.7 ± 15.9) were not significantly different in respect to values at month 12 (41.2 ± 11.18). At 24 months, only 1 out 19 patients had a value <20 ng/mL. There was a significant decrease with time of mean values of bALP (*p* < 0.0216), with no significant changes between 12 and 24 months. No significant changes were observed as far as ionised calcium or creatinine were concerned. Conclusions: The long-term administration of calcifediol maintains stable and sustained 25(OH)D concentrations, with no safety concerns.

## 1. Introduction

Vitamin D insufficiency (and deficiency) is largely prevalent in the world [1]. If we consider a threshold of 12 ng/mL (30 nmol/L; to convert ng/mL to nmol/L multiply the ng/mL by 2.5), it has been estimated that about 5% of the American population and about 9% of the Canadian population have serum values of 25(OH)D below this level [2]. Furthermore, the risk of being vitamin D-deficient in the United States is higher in the black population. Based on the same thresholds, about 13% of the population is at risk of vitamin D deficiency in Europe. This percentage can increase up to 28–65% if we take into consideration ethnic minority groups within Northern European countries [2]. However, the prevalence of serum 25(OH)D below 20 ng/mL raises in the range of 27.2 to 61.4%. Considering both continents, it has been estimated that 120 million subjects are vitamin D-deficient using a 12 ng/mL threshold, while there are 390 million subjects with 25(OH)D below 20 nmol/L. These figures are astonishing and are a call to action for specific initiatives to eliminate this “global pandemic”. However, even though many such initiatives have been undertaken over the last few years to increase awareness about this clinical problem, the situation seems to be unchanged.

Low values of 25(OH)D, considered the best biological marker of body stores of vitamin D from all sources, have detrimental effects on the skeleton and have been also associated with a number of non-skeletal outcomes [3,4,5]. Indeed, a huge number of epidemiological studies have consistently demonstrated that low 25(OH)D values are associated with an increased risk of cancer and chronic diseases, among which are cardiovascular, neurological, autoimmune and infectious diseases. Some studies have also found an association with an increased risk of mortality [6].

Vitamin D is synthesised from 7-dehydrocholesterol in the skin by absorbing ultraviolet B (UVB) radiation. Vitamin D can also be obtained from the diet or by means of dietary supplementation in the form of either vitamin D3 (cholecalciferol) or D2 (ergocalciferol). Within a few hours of being synthesised in the skin or taken with foods, it undergoes two consecutive hydroxylation steps: the first one is represented by the addition of one hydroxyl group in position 25 to generate 25(OH)D in the liver; the second step involves the addition of another hydroxyl group in position 1 in the kidney to generate the final metabolite, that is, 1,25(OH)_2_D. The most important function of the final metabolite is an effect on intestinal calcium absorption. Without vitamin D, only 10–15% of dietary calcium and about 60% of phosphorus are absorbed. Vitamin D sufficiency enhances calcium and phosphorus absorption by 30–40% and 80%, respectively. Other tissues contain the enzymes needed for hydroxylation, but the liver is the main source for 25-hydroxylation, and the kidney is the main source for 1α-hydroxylation. Control of the metabolism of vitamin D to its active metabolite, 1,25(OH)_2_D, is exerted primarily at the renal level, where calcium, phosphorus, parathyroid hormone, fibroblast growth factor 23 (FGF23) and 1,25(OH)_2_D regulate the levels of 1,25(OH)_2_D that are produced.

There are essentially four possibilities for increasing circulating levels of 25(OH)D, each carrying inherent advantages and disadvantages. Firstly, by increasing skin synthesis of vitamin D with exposure to UVB light. Secondly, by increasing the intake of vitamin D foods; however, few foods contain vitamin D. The flesh of fatty fish (that is, trout, salmon, tuna and mackerel) and fish oils are among the best sources. Beef liver, egg yolks and cheese have small amounts of vitamin D, primarily in the form of vitamin D3 and its metabolite 25(OH)D3. Mushrooms provide variable amounts of vitamin D2. Thirdly, by enriching food with vitamin D or 25(OH)D and finally, by pharmacological supplementation [7]. Concerning the last option, there are a number of commercial products that can be utilised [8]; they basically recapitulate the biosynthetic pathway of vitamin D from its origin to the final metabolite. The most utilised of these products are ergocalciferol, cholecalciferol, calcifediol, 1α(OH)D and calcitriol, even though their utilisation is markedly different in European and American countries. Other compounds are utilised in specific clinical conditions, such as in chronic kidney disease [9].

Calcifediol is the intermediate metabolite in the metabolic pathway that leads from cholecalciferol to calcitriol production, being formed by the addition of a 25-hydroxyl group to its precursor (mainly by the enzymes microsomal CYP2R1 and mitochondrial CYP27A1). Calcifediol binds to Vitamin D-binding protein in the circulation. It has distinct metabolic and pharmacokinetics profiles, compared to both its precursor and downstream final metabolite [10]. A number of previous investigations have shown that calcifediol administration determines a rapid increase of circulating 25(OH)D compared to cholecalciferol [11]. Indeed, calcifediol is rapidly absorbed by intestinal cells and transported through the portal vein, and therefore, it is immediately available in the circulation. Native vitamin D is transported more slowly by chylomicrons via the lymphatic system. The rapid absorption and its independence from hepatic 25-hydroxylation lead to pharmacokinetics that make calcifediol an immediately available substrate for the synthesis of calcitriol, either for systemic transport or for local paracrine or autocrine actions. In this context, Charoenngam et al. [11] evaluated the effect of cholecalciferol and calcifediol in 6 patients with malabsorption and 10 healthy subjects, reporting thought-provoking pharmacokinetic data. After administration of 900 mcg of cholecalciferol, the area under the curve was 68% lower for patients with malabsorption than for healthy patients. However, the administration of 900 mcg of calcifediol revealed no differences between the two groups. This suggests that the bioavailability of 900 mcg of orally administered 25(OH)D_3_ was not different between malabsorptive patients and healthy participants.

Concerning long-term effects, the longest time frame of calcifediol investigation has been 12 months, with studies differing in both the amount and schedule of administration [12,13,14]. In the longest-lasting study to date, the monthly administration of 0.266 mg of calcifediol for as long as one year produced stable and sustained 25(OH)D concentrations with no associated safety concerns [12].

This study has been therefore carried out to investigate the 2-year effect of monthly 0.266 mg calcifediol administration in restoring and maintaining 25(OH)D levels in postmenopausal females with hypovitaminosis D.

## 2. Materials and Methods

This is an open-label study in which patients received calcifediol at a dose of 0.266 mg monthly (Neodidro soft gelatine capsule, Bruno Farmaceutici S. p. A., Rome, Italy) for a period of two years. In addition to being a response to the lack of long-term data (i.e., more than one year), this schedule was chosen because it is the one suggested by the Italian Medical Agency (note 96 of the Agenzia Italiana del Farmaco, AIFA, Rome, Italy) for vitamin D replenishment when circulating values are between 13 and 20 ng/mL or above 20 ng/mL. Furthermore, this type of administration aims, on the one hand, at avoiding any serious adverse events while, on the other hand, guaranteeing a good adherence in respect to daily or weekly administration. All participants signed their informed consent prior to entering the investigation.

We initially enrolled 45 consecutive postmenopausal patients with total 25(OH)D values less than 30 ng/mL, during the period September 2019–September 2020. Being postmenopausal was defined as amenorrhea for more than 6 months or follicle-stimulating hormone levels above 30 I. U./L, with oestradiol values less than 30 pg/mL) They attended our mineral metabolism unit because they had already carried out a bone mineral density (BMD) measurement or because they were worried about a possible diagnosis of osteoporosis. Exclusion criteria were medical or therapeutic conditions that could interfere with our investigation. These were represented by the following: progressive major illnesses, severe malabsorption syndrome, stage IV chronic kidney disease (that is, an estimated glomerular filtration rate between 15 and 29 mL/min, together with moderate to severe kidney damage), nephrolithiasis, Paget’s disease of the bone, primary hyperparathyroidism, genetic and acquired hypoparathyroidism, hypercalcaemia, hypothyroidism and hyperthyroidism, sarcoidosis, hypercalciuria and intolerance to calcifediol. Patients taking treatments interfering with bone and mineral metabolism such as glucocorticoids, diuretics, lithium, immunosuppressants, antiretroviral therapy and other drugs possibly interfering with vitamin D absorption and catabolism were also excluded. The initial visit included a physical examination, blood collection and BMD imaging of the spine and hip with a vertebral fracture assessment, as previously described [15]. An experienced skeletal radiologist evaluated the presence of fractures according to an algorithm-based qualitative approach. Both the vertebral shape and the appearance of the endplate were evaluated, to clearly differentiate vertebral fractures from developmental variants, degenerative change and so on. Patients were therefore instructed to return at 3, 6, 9, 12 and 24 months for measuring serum total 25(OH)D, ionised calcium, creatinine and isoenzymes of alkaline phosphatase (bALP). Professional nurses obtained blood samples at each visit and carried out a medical history for possible intercurrent illnesses, changes in the usual therapy and possible side effects.

### 2.1. Biochemical Analysis

Briefly, blood samples were taken between 8.00 and 10.00 a.m. for the measurement of serum ionised calcium (Ca^++^), 25-hydroxyvitamin D, creatinine and isoenzymes of alkaline phosphatase in the fasting state. Details for the measurement of blood chemistries and bone mineral density have been previously described [16,17]. More specifically, except for Ca^++^, the level of which was measured immediately after collection (within 2 h of sampling, while the room temperature was kept at 4 °C), aliquoted blood samples were stored at −80 °C and assayed at a later time in batches. Serum Ca^++^ determination was carried out using an ion-selective electrode with the fully automated biochemical analyser NOVA 8 (Nova Biomedical, Waltham, MA, USA); 25(OH)D was measured by chemiluminescence–immunoassay (CLIA) with the fully automated LIAISON^®^ analyser (Diasorin S.p.A. Saluggia, Italy). Due to technical limitations imposed by use of the LIAISON^®^ instrument, the 25(OH)D assays were performed on a quarterly basis. The determinations of serum 25(OH)D were performed with a competitive one-step backfill chemiluminescence assay (Vitamin D TOTAL Assay, DiaSorin USA, Stillwater, MN, USA) having a measurement range of 4–150 ng/mL and functional sensitivity ≤4.0 ng/mL; the intra- and inter-assay precision were 8.9% and 12.8%, respectively, with a reported 100% detection of both 25-hydroxyvitamin D2 and 25-hydroxyvitamin D3. Serum creatinine was measured by a Cobas c501 analyser (Cobas^®^ Roche Diagnostics S.p.A., Monza, Italy). The bone isoenzyme of alkaline phosphatase was measured by IDS-iSYS Ostase^®^ BAP assay (Immunodiagnostics Systems, Boldon, UK).

Safety and tolerability were also examined throughout the study. This study was performed in accordance with the Declaration of Helsinki, as well as local laws and regulations. The ethical committee of our university reviewed and approved the protocol.

### 2.2. Statistical Analysis

Quantitative variables are summarised by mean ± standard deviation. Student’s *t* test or the Mann–Whitney test was used for pairwise comparisons as appropriate, depending on the characteristics of the data. The effect of time on the trend of each single outcome has been assessed using linear mixed-effect models for repeated measures. Post hoc pairwise comparisons were performed to assess the significance of the differences of the outcomes at various times. For statistical significance, a *p* value < 0.05 was considered appropriate.

## 3. Results

We initially enrolled 45 consecutive postmenopausal patients attending our mineral metabolism service who satisfied our inclusion and exclusion criteria, with serum 25(OH)D levels below 30 ng/mL. Thirty-six of them initially agreed to participate. The enrolment period (September 2019–September 2020) and the follow-up observation period coincided with the beginning of the COVID-19 pandemic in Italy and subsequent restrictions on mobility. In respect to the initial protocol, this had two important consequences: first, not all the enrolled patients ended the protocol, and secondly, not all patients adhered to the scheduled visits, mainly because they were afraid of coming to our hospital, which was a referral centre for COVID-19. We therefore report the results according to a per-protocol analysis.

Table 1 shows the anthropometric, biochemical and densitometric parameters of the 36 patients who initially adhered to the study protocol. There were no significant statistical differences in respect to those that declined to participate.

The mean (±1 SD) age of the patients was 62.4 ± 9 years and about 18% of them had one or more vertebral fractures. Basal total 25(OH)D levels were 20.9 ± 4.2 ng/mL. Two patients were already taking generic alendronate (without vitamin D), and a further nine patients started taking various therapies for osteoporosis during the study, owing to fractures or a high risk of fractures as determined by FRAX^®^. The mean T-score values at the three sites considered (that is, lumbar spine. femoral neck and total hip) were in the osteopenic range. Twelve patients were treated for essential hypertension, two for type 2 diabetes mellitus and nine for hyperlipidaemia. None of them was taking drugs that could interfere with vitamin D metabolism.

There was a continuous increase in mean 25(OH)D values (*p* for trend < 0.001) during the 2-year observation study. However, the mean values at month 24 (36.7 ± 15.9) were not significantly different in respect to values at month 12 (41.2 ± 11.18) (Figure 1). At the end of the study (i.e., 24 months) only 1 out 19 patients had a value less than 20 ng/mL, and 5 patients had values above 20 ng mL.

There was a significant decrease with time in mean values of bone ALP (*p* < 0.0216), with no significant changes in mean values between 12 and 24 months (22.9 ± 11.9 vs. 20.8 ± 20.5 mg/L) (Figure 2).

No significant changes were observed as far as mean values of serum ionised calcium (Figure 3) or creatinine were concerned. No serum ionised calcium above 1.33 nmol/L was documented.

Throughout the entire period of treatment, 12 of the patients enrolled reported at least one adverse event (A. E.). Only one treatment-related A. E. was reported by one patient (mild dyspepsia) that in any case did not force her to withdraw from the investigation. Two patients developed a COVID-19 infection during the study. No other safety issues were associated with the present analysis; in particular, there were no deaths and no serious adverse events attributable to calcifediol. The maximum 25(OH)D value reached in the entire population considered for the whole study period was 79.7 ng/mL at month 24, in a patient whose basal value was 26.3 ng/mL.

## 4. Discussion

To the best of our knowledge, this is the first open-label investigation carried out to evaluate the efficacy and safety of monthly 0.266 mg calcifediol administered for as long as 2 years. Indeed, no previous investigation in the English-language literature has covered such a long period of time with a monthly or other schedule of calcifediol administration. For example, Bischoff-Ferrari and coworkers [4] performed an investigation in 10 healthy postmenopausal women by administering 20 mcg of calcifediol per day over 4 months. They showed that this amount and schedule of administration resulted in a safe, immediate and sustained increase in 25(OH)D serum levels in all participants. Russo and coworkers [12] investigated the effect of the monthly oral administration of 500 mcg of calcifediol in 18 normal women aged 24–72 years for a period of four months. They found that the pulsed monthly administration of 500 mcg of calcifediol can be considered an alternative for vitamin D repletion in respect to cholecalciferol, especially in conditions in which a rapid normalisation of 25(OH)D levels is needed. Graeff-Armas and coworkers [18] administered three different calcifediol doses (10, 15 and 20 mcg daily) to men and postmenopausal women aged 50 years or older, for six months. They essentially demonstrated that 25(OH)D_3_ is very effective in raising 25(OH)D concentrations. Furthermore, they showed that once supplementation is discontinued, the elimination rate of 25(OH)D_3_ is faster than following the suspension of cholecalciferol, similarly to what has been subsequently reported by Perez-Castrillon and coworkers [14]. Minisola and coworkers [13] enrolled 87 Caucasian, community-dwelling, postmenopausal women, aged 55 years or older, with vitamin D levels below 30 ng/mL. These subjects were randomised to receive three different dosages of calcifediol: 20 mcg daily, 40 mcg daily or 125 mcg weekly for 3 months. They demonstrated that different daily or weekly dosages of calcifediol have a mid-term efficacy and safety on the main parameters of mineral metabolism, when used to treat vitamin D inadequacy or deficiency in this category of subjects. Navarro Valverde et al. enrolled 30 postmenopausal osteoporotic women with an average age of 67 years who were vitamin D-deficient. Calcifediol was administered to three groups of 10 people each, at a dose of 20 mcg daily, 266 mcg weekly or every other week, respectively, for 12 months. At the end of the study, levels above 20 ng/mL were attained by all the subjects, independently of the schedule of administration [19]. Shieh and coworkers [20] enrolled 19 subjects (mean age 34.8 ± 8.6 years), to whom they administered 20 mcg calcifediol daily. They found that calcifediol rapidly and robustly increases serum concentrations of both total and free 25(OH)D and more reliably increases circulating 25(OH)D to levels of 30 ng/mL by the fourth week of supplementation. Gonnelli and coworkers [21] randomised fifty osteopenic/osteoporotic women with serum levels of 25-hydroxyvitamin D between 10 and 20 ng/mL to a 6-month treatment with oral calcifediol 20 mcg/day (*n* = 23, mean age 62.4 ± 7.4 years) or oral calcifediol 30 mcg/day (*n* = 25, mean age: 61.5 ± 8.3 years). Calcifediol was able to rapidly normalise the vitamin D deficiency; furthermore, they suggested that the 30 mcg daily dose could be utilised in those patients who need to rapidly reach optimal 25OHD levels. Ruggiero and coworkers administered to community-dwelling women and men, older than 75 years, who were consecutively admitted to a geriatric acute care ward, a weekly dose of 150 mcg of calcifediol. They found that the administration of calcifediol is beneficial in oldest-old people since it rapidly increased serum levels and reaches the optimal target threshold, especially among those with a comorbidity, taking multiple drugs and showing low muscle strength [22]. Finally, Pérez-Castrillòn and collaborators [14] administered 0.266 mg monthly calcifediol for 12 months to 58 postmenopausal women (aged 64.3 ± 8.2 years) with 25(OH)D levels lower than 20 ng/mL. Treatment for one year produced stable and sustained 25(OH)D concentrations with no associated safety concerns.

Fron a review of the previous literature, it is apparent that our investigation fills a gap. Indeed, the 0.266 mg monthly dose of calcifediol has never been previously utilised, with only one notable exception. However, the investigation of Pérez-Castrillòn and coworkers did not have the same duration as the one we carried out. Furthermore, it is also apparent that, unless the reason underlying vitamin D insufficiency is removed, treatment to correct a condition of hypovitaminosis D should be lifelong. Having carried out a study of such a long duration (i.e., 24 months) represents another step forward.

Our data are in line with previous investigations, in that they show the favourable and quick effect of calcifediol in restoring 25(OH)D values to the threshold of sufficiency, regardless of how this is settled by various scientific societies. In this respect, it is important to highlight that, considering the time course of 25(OH)D values following monthly calcifediol administration (Figure 1), starting from the third month of observation, all mean values reached the threshold of 30 ng/mL. The time points of our protocol did not include earlier serum 25(OH)D measurements. However, we previously showed that serum 25(OH)D values significantly and promptly rose and plateaued above the 30 ng/mL threshold, remaining within the safety interval after 14 days of treatment, with a similar efficacy for daily and weekly dose regimens. The different dosages were also equally effective in controlling secondary hyperparathyroidism.

There are some clinical conditions in which a rapid normalisation of 25(OH)D levels is desirable. In this context, it has been shown the low levels of vitamin D might be responsible for the acute-phase reaction induced by the intravenous infusion of amino bisphosphonate and, rarely, also, when it is administered by the oral route. Bertoldo and coworkers showed that patients with 25(OH)D levels below 30 ng/mL have a threefold increased risk of acute-phase reaction associated with zoledronic acid infusion [23]. Even though the mechanisms by which vitamin D status can influence the appearance of acute-phase reaction have not been completely elucidated until now, the most widely accepted mechanism underscores a direct effect on delta γδ T cells. Indeed, this is the population of T cells mainly involved in the acute-phase reaction. The attainment of vitamin D sufficiency by calcifediol administration in a short period of time is an appealing strategy to avoid this untoward side effect. In fact, it has been proven that the mean increase in serum 25(OH)D after the intake of 1 mcg cholecalciferol is 1.53 nmol/L, whereas the mean increase was 4.76 nmol/L with calcifediol [24]. Next, it has been shown that, in order that drugs utilised for osteoporosis fully exert their effect, patients should be vitamin D replete [25]. We now have available to us a number of drugs (mainly anabolic) that exert their effect in the first few months after initial treatment, as evidenced by Kaplan–Meier estimates in respect to placebo or antiresorptive agents. This rapid effect is especially important in those patients who have sustained a recent fracture. Therefore, a condition of vitamin D sufficiency is a prerequisite to avoid the deleterious effects of secondary hyperparathyroidism. Finally, adequate intakes of calcium and vitamin D are recommended when taking drugs (i.e., denosumab) that heavily suppress osteoclast activity, thus limiting the body’s ability to mobilise calcium from bone. In these conditions, the administration of vitamin D with a rapid effect on calcium absorption is mandatory, in order to reduce the risk of treatment-induced hypocalcaemia. In all previous clinical conditions, the administration of calcifediol can be envisaged as a good strategy to allow patients to start treatment immediately. Indeed, the administration of calcifediol has predictable effects on the increase in metabolite values not being dependent on the dose administered. Then, epigenetic factors have a minor role, because the metabolite we measure is the one we administer.

The most important finding of our investigation is represented by the long-term stabilisation of 25(OH)D levels during the whole time frame of our investigation (i.e., 2 years). Indeed, while there was a trend towards increase in the first year of treatment, the levels then plateaued, so that there were no statistically significant differences between mean values obtained at 12 and 24 months of observation. Our results mirror those of previous authors using similar amounts and schedules of calcifediol administration [14]. They showed persistent stable 25(OH)D levels, after reaching the maximum value at 4 months, for the remaining period of 8 months (however, the full length of the study was only one year). Our findings also suggest that, after reaching a steady state, there is no further increase in 25(OH)D values, most probably due to a fine-tuning of CYP24A1. This enzyme is of major clinical and physiologic importance, serving to regulate the catabolism of both 25(OH)D and 1,25(OH)_2_D, thus avoiding excessive production of the final metabolite of the vitamin D biosynthetic pathway.

We also demonstrated a significant trend toward a decrease in mean values of isoenzymes of alkaline phosphatase, even though, also in this case, the mean values at 12- and 24-month observation were not significantly different. This finding can be interpreted in the light of the inhibitory action of calcifediol on parathyroid hormone secretion, with a consequent decrease in bone turnover. These findings are in line with our previous observations, in which patients were treated with a monthly administration of 500 mcg of calcifediol for a period of 4 months [12]. Unfortunately, not in all studies carried out so far to evaluate the efficacy of calcifediol have markers of bone turnover been assayed. Future studies with measurement of both formation and resorption markers are needed to fill this gap.

Toxicity is an infrequent event in all the studies published so far with calcifediol. In the context of safety, it should be highlighted that safety margins are quite broad, independently of whether we utilise cholecalciferol or its derivatives. In addition, vitamin D toxicity with the classical biochemical (i.e., hypercalcaemia, hypercalciuria, decrease in glomerular filtration rate) and biochemical (i.e., nephrocalcinosis) manifestations is a rare finding, when patients adhere to the prescribed dose of the compound. Most published cases of vitamin intoxication so far reported in the literature indeed arise from self-administering huge doses of vitamin D. It is reassuring that no blood levels above 150 ng/mL have been reported in previous investigations. According to these previous findings, the maximum value of 25(OH)D detected in our investigation was 79.7 ng/mL. It can be hypothesised that extremely high levels of 25(OH)D can be reached if there is a concurrent clinical condition, such as an associated disease that increases the synthesis of 1,25-dihydroxycholecalciferol or genetic defects that alter the metabolism of vitamin D by reducing the concentration of 24,25-dihydroxycholecalciferol, a safety mechanism that prevents toxicity. The main biochemical manifestations of toxicity are represented by the finding of hypercalcaemia and hypercalciuria, together with reduced kidney function. It is worthwhile to emphasise that we did not detect any episode of hypercalcaemia during the entire observation period. This finding, together with the lack of significant changes of serum creatinine and no reported side effects, argue in favour of the safety and tolerability of monthly calcifediol administration. In the biggest randomised controlled trial [14], comparing the efficacy and safety of calcifediol 0.266 mg soft capsules with cholecalciferol in 303 postmenopausal women, no deaths or serious adverse events were reported. Furthermore, the maximum 25(OH) D level reached by a patient was 64.4 ng/mL during the first month of treatment due to a medication error (a weekly instead of monthly intake of calcifediol). The withdrawal criterion for 25(OH)D levels in this clinical trial was a value greater than 80 ng/mL; it is important to highlight that no patient reached that threshold. Also in that study, no clinically relevant hypercalcaemia cases were reported in any of the patients treated with calcifediol. Finally, the maximum calcium level reached was 11.1 mg/dL in one patient, whereas the remaining patients had levels lower than 10.8 mg/dL. Taken together, these data indicate that the finding of an increased serum calcium value is infrequent with a correct administration of 0.266 mg monthly of calcifediol. In our investigation, we utilised a monthly dose of 0.266 mg of calcifediol. However, we are aware of the possibility that such a dose can be increased in specific conditions (i.e., when initially treating a severely deficient patient with vitamin D values less than 12 ng/mL) or when higher thresholds may be desirable (i.e., when treating autoimmune diseases such as rheumatoid arthritis). In these previous conditions, more frequent laboratory tests are advisable, both to check threshold achievement and also the lack of biochemical measures of toxicity. Theoretically, a fine-tuning of the amount of vitamin D to be administered can be envisaged if basal 25(OH)D values are available.

We did not report urinary calcium excretion, because the patients enrolled were not willing to have a troublesome urinary collection in the middle of the COVID-19 pandemic. Urinary collection was performed in a few patients, which did not allow us to have the statistical power to draw meaningful conclusions.

Several limitations of this study should be considered. The first one is represented by the per-protocol, and not intentional, analysis of the data obtained. However, this approach was justified by the peculiar period in which this investigation was carried out (i.e., in the midst of the COVID-19 pandemic) that did not allow a full adherence to the initial scheduled visits of the protocol. In this context should also be interpreted the lack of the inclusion of a placebo control group. Secondly, a full evaluation of the biochemical parameters of skeletal metabolism would have been desirable. However, our focus was the attainment of 25(OH)D values in the sufficiency range in the long term without reporting any serious adverse events. This primary endpoint, not previously targeted in previous investigations, was reached in our study. Finally, we did not include a group treated with cholecalciferol, since there is an abundance of papers showing the different metabolism and predictability of efficacy of calcifediol in respect to cholecalciferol, so that a further study addressing this problem would have resulted in useless repetition.

The main strength of our protocol derives from the homogeneity of the population studied from a single centre and, most importantly, the duration of the study—that is, 24 months—never reached in previous studies published in the English-language literature. Another particular aspect that should be emphasised is represented by the formulation of calcifediol, together with the schedule utilised. Taking one soft gelatine capsule per month is easier compared to taking a certain number of drops each day, as reported in the majority of studies previously cited. This kind of formulation is available both in Italy and Spain, even though recommendations concerning doses may differ (based on initial vitamin D values in Italy or on the characteristics of the population in Spain). The desired final effect on vitamin D status is reached by utilising this formulation, as can be observed by the stable circulating levels of 25(OH)D throughout the whole observation period.

## 5. Conclusions

In conclusion, this is the first open-label investigation carried out for as long as 2 years with monthly 0.266 mg calcifediol administration. The long-term administration of calcifediol maintained stable and sustained 25(OH)D concentrations. Most importantly, no safety concerns were reported by the patients.

## Figures and Tables

**Figure 1 nutrients-16-01754-f001:**
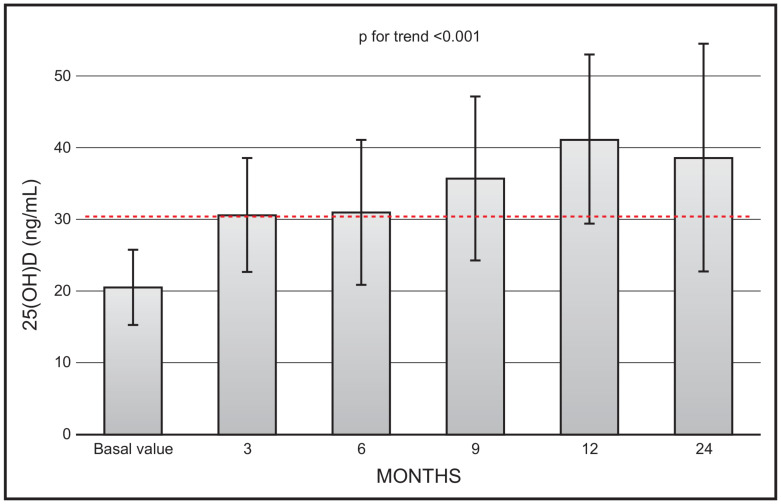
Time course of serum 25(OH)D following monthly administration of 0.266 mg calcifediol. Error bars indicate 1 SD.

**Figure 2 nutrients-16-01754-f002:**
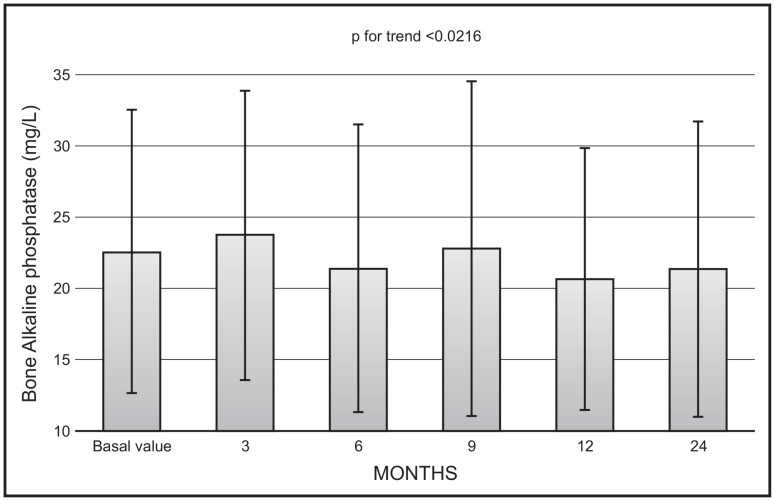
Time course of serum bone alkaline phosphatase following monthly administration of 0.266 mg calcifediol. Error bars indicate 1 SD.

**Figure 3 nutrients-16-01754-f003:**
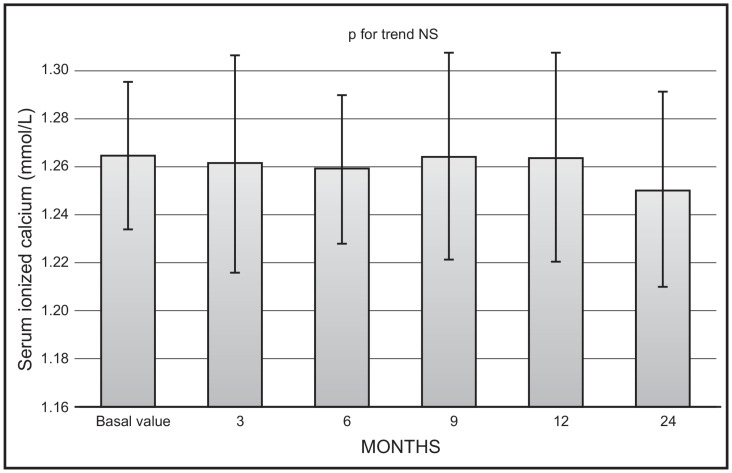
Time course of serum ionised calcium following monthly administration of 0.266 mg calcifediol. Error bars indicate 1 SD.

**Table 1 nutrients-16-01754-t001:** Main anthropometric, biochemical and densitometric parameters of patients investigated.

**Age (years)**	62.4 (±9)
**Weight (kg)**	60.7 (±9.5)
**Height (cm)**	161.8 (±7.1)
**Body Mass Index (kg/m** ** ^2^ ** **)**	23.1 (±3.4)
**Years since menopause**	14 (±8.7)
**Vertebral fractures**	17.6%
**L1–L4 BMD (g/cm** ** ^2^ ** **)**	0.840 (±0.161)
**L1–L4 T-score (SD)**	−2.0 (±1.3)
**Femoral neck BMD (g/cm^2^)**	0.654 (±0.102)
**Femoral neck T-score (SD)**	−1.92 (±0.86)
**Femoral total BMD (g/cm^2^)**	0.765 (±0.124)
**Femoral total T-score (SD)**	−1.48 (±1.05)
**25 OH vitamina D (ng/mL)**	20.5 (±5.3)
**Ionised calcium (mmol/L)**	1.26 (±0.03)
**Total calcium (mg/dL)**	9.2 (±0.7)
**Phosphate (mg/dL)**	3.7 (±0.3)
**Creatinine (mg/dL)**	0.74 (±0.17)
**Bone alkaline phosphatase (µg/L)**	22.9 (±10.4)

## Data Availability

Some or all datasets generated during the current study are not publicly available due to privacy but are available from the corresponding author upon reasonable request.

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
