# Peer review of "Effect of 2 Years of Monthly Calcifediol Administration in Postmenopausal Women with Vitamin D Insufficiency"

_nutrients, 2024, doi:10.3390/nu16111754_

Round 1
Reviewer 1 Report
Comments and Suggestions for Authors
The authors prospectively analyzed the effectiveness and safety of administering calcifediol monthly for 2 years. They reported a significant increase in 25(OH)D levels over the treatment period with no observed adverse effects. While the study holds clinical significance, several limitations need addressing.
1) Lack of an adequate control group and the small sample size limit the study's robustness. A comparison with the commonly used cholecalciferol would enhance its clinical relevance.
2) Rationale behind choosing a monthly formulation of calcifediol and its clinical significance require clarification.
3) Line 122: could you please elaborate on the definition of "stage IV chronic kidney disease"?
4) Please provide an information of the monthly calcifediol formulation administered in this study.
5) Please provide the approved IRB number for transparency.
6) Lines 220-264 contain an excessive amount of information on existing literature without highlighting how this study differs from previous ones. Focusing on the distinctions between this study and others with similar formulations, such as why the monthly regimen and the specific dosage were chosen, would improve clarity.
7) Line 266, what is the rationale behind the observed quick effect? While this study showed an outcome within 3 months after the first dose, IV cholecalciferol formulations typically exhibit an immediate rise in 25(OH)D levels, known as a quick effect.
8) Overall, the discussion section appears lengthy. While the authors emphasize the study's main strength—the two-year duration—a more detailed discussion on the formulations used would be beneficial. Specifically, comparing and contrasting with other doses would enhance the discussion's depth.
Author Response
The authors prospectively analyzed the effectiveness and safety of administering calcifediol monthly for 2 years. They reported a significant increase in 25(OH)D levels over the treatment period with no observed adverse effects. While the study holds clinical significance, several limitations need addressing.
We thank the reviewer for his positive general comment. As you can see in the following points, we addressed the criticism you raised so that the paper is much improved in respect to the original submission.
1) Lack of an adequate control group and the small sample size limit the study's robustness. A comparison with the commonly used cholecalciferol would enhance its clinical relevance.
We thank the reviewer for this observation that now is included among the limits of the study. Concerning the second point, we believe that the literature is plenty of papers showing different metabolism and predictability of efficacy of calcifediol in respect to cholecalciferol so that a further study addressing this problem would have resulted in a useless repetition.
2) Rationale behind choosing a monthly formulation of calcifediol and its clinical significance require clarification.
The choice of administering 0.266 mg of calcifediol monthly is in line with our previous research aimed at finding the right dose of this Vitamin D derivative that from one side could avoid any serious adverse event while, on the other end, guarantee a good adherence in respect to daily or weekly administration. Secondly, no previous long-term studies (i. e., 2 years) have been carried out with such a regimen; this point has been already addressed in the manuscript. Finally, and most importantly, this is the schedule suggested by Italian Medical Agency (note 96 of the Agenzia Italiana del Farmaco, AIFA) for vitamin D replenishment when circulating values are between 13 and 20 ng/mL or above 20 ng/ml.
Some of these explanations have been added in the text.
3) Line 122: could you please elaborate on the definition of "stage IV chronic kidney disease"?
The KIDGO definition has been added in brackets.
4) Please provide an information of the monthly calcifediol formulation administered in this study.
This information was already present at the initial of headlines “Materials and Methods”, that is, calcifediol at a dose of 0.266 mg monthly (Neodidro soft gelatine capsule, Bruno Farmaceutici S. p. A., Rome, Italy).
5) Please provide the approved IRB number for transparency.
Numbers and dates have been added.
6) Lines 220-264 contain an excessive amount of information on existing literature without highlighting how this study differs from previous ones. Focusing on the distinctions between this study and others with similar formulations, such as why the monthly regimen and the specific dosage were chosen, would improve clarity.
Many thanks for highlighting this important point. We believe that recapitulate previous literature in the field is important for those that are not familiar with. However, we agree with you that emphasizing differences in respect to these studies is important. Therefore, a new paragraph has been added, following your suggestion.
7) Line 266, what is the rationale behind the observed quick effect? While this study showed an outcome within 3 months after the first dose, IV cholecalciferol formulations typically exhibit an immediate rise in 25(OH)D levels, known as a quick effect.
Thanks for rising this point. We agree with you that the first observation was obtained at 3 months. However, our statement of a “quick effect” was based on the finding of an observed mean value of 30 ng/mL, that is the threshold of sufficiency acknowledged by all scientific societies. Once again, our findings are in line with previous literature on this field. Regarding intravenous formulation of cholecalciferol we are unaware of such studies but we are willing to include once they are provided.
8) Overall, the discussion section appears lengthy. While the authors emphasize the study's main strength—the two-year duration—a more detailed discussion on the formulations used would be beneficial. Specifically, comparing and contrasting with other doses would enhance the discussion's depth.
Your point has been taken. A phrase concerning formulations has been added also in comparison with other doses reported in the literature.
We hope that having satisfied all the issue raised, you can now consider the paper suitable for definitive publication.
Reviewer 2 Report
Comments and Suggestions for Authors
The authors assessed the effects of a long-term supplementation of calcifediol (0.266 mg) of 24 months on the correction and maintenance of vitamin D status in postmenopausal women, and reported the administration of calcifediol maintained stable and sustained 25(OH)D concentrations with no safety concerns. The study provides some meaningful findings concerning the use of calcifediol to maintain a suitable vitamin D status in the body. However, the quality and significance of the study in limited.
Major:
Although the authors stated there are certain objective obstacles in performing the study, such as Covid 19, the data should support the conclusions. However, the indexes detected in the present study is limited, with not sufficient information to the reader.
Also, as to the sample size, the authors initially enrolled 45 postmenopausal women, and then 36 of them agreed to participate. Is the sample size enough to meet the efficiency of statistical analysis?
The discussion should be improved to a large extent.
Minor:
The SD for ALP is very large. How did the author analyze the data?
Comments on the Quality of English Language
Minor editing of English language is required.
Author Response
The authors assessed the effects of a long-term supplementation of calcifediol (0.266 mg) of 24 months on the correction and maintenance of vitamin D status in postmenopausal women, and reported the administration of calcifediol maintained stable and sustained 25(OH)D concentrations with no safety concerns. The study provides some meaningful findings concerning the use of calcifediol to maintain a suitable vitamin D status in the body. However, the quality and significance of the study in limited.
We thank the reviewer for his positive comments. We tried to improve the quality and significance of the manuscript, at our best. Thanks to the criticism you raised, we now believe that the paper is much improved in respect to the original submission.
Major:
Although the authors stated there are certain objective obstacles in performing the study, such as Covid 19, the data should support the conclusions. However, the indexes detected in the present study is limited, with not sufficient information to the reader.
We thank the reviewer for this observation. However, we believe that there is not an overinterpretation of the results obtained throughout the paper. We have been very focused on the level of 25(OH)D during the entire observation period and on the safety issues. We are willing to modify the text, where this overinterpretations occurred.
Also, as to the sample size, the authors initially enrolled 45 postmenopausal women, and then 36 of them agreed to participate. Is the sample size enough to meet the efficiency of statistical analysis?
Thank you for your observation. While the sample size may appear small, this is a longitudinal study, therefore the reduced sample size is offset by repeated observations of the same patients over multiple occasions. For instance, the power calculation for a paired t-test with a Cohen’s d of 0.5 over two occasions with a sample size of n=36 is 0.83. For a repeated measures ANOVA over five occasions and the same sample size, the power exceeds 0.99.
We tried to improve the discussion, according to your suggestion, by adding new phrases to better explain specific aspects.
Minor:
The SD for ALP is very large. How did the author analyze the data?
Data have been analysed using a mixed effect model for repeated measures. Even if the variability may seem large, observing the same patient over time improves the power of the analysis.
We hope that having satisfied all the issue raised, you can now consider the paper suitable for definitive publication.
Round 2
Reviewer 1 Report
Comments and Suggestions for Authors
The authors have adequately addressed the issues raised.
Reviewer 2 Report
Comments and Suggestions for Authors
The paper has been improved. However, still some flaws including spelling error, symbol, etc. should be checked carefully by the authors.